# Maternal but Not Paternal High-Fat Diet (HFD) Exposure at Conception Predisposes for ‘Diabesity’ in Offspring Generations

**DOI:** 10.3390/ijerph17124229

**Published:** 2020-06-13

**Authors:** Karen Schellong, Kerstin Melchior, Thomas Ziska, Rebecca C. Rancourt, Wolfgang Henrich, Andreas Plagemann

**Affiliations:** 1Division of ‘Experimental Obstetrics’, Clinic of Obstetrics, Charité–Universitätsmedizin Berlin, corporate member of Freie Universität Berlin, Humboldt-Universität zu Berlin, and Berlin Institute of Health, Campus Virchow-Klinikum, 13353 Berlin, Germany; karen.schellong@charite.de (K.S.); kerstin.melchior@charite.de (K.M.); thomas.ziska@charite.de (T.Z.); rebecca.rancourt@charite.de (R.C.R.); 2Clinic of Obstetrics, Charité–Universitätsmedizin Berlin, corporate member of Freie Universität Berlin, Humboldt-Universität zu Berlin, and Berlin Institute of Health, Campus Virchow-Klinikum, 13353 Berlin, Germany; wolfgang.henrich@charite.de

**Keywords:** developmental/perinatal programming, maternal and paternal overnutrition, high-fat diet, intergenerational effects, obesity, diabetes

## Abstract

While environmental epigenetics mainly focuses on xenobiotic endocrine disruptors, dietary composition might be one of the most important environmental exposures for epigenetic modifications, perhaps even for offspring generations. We performed a large-scale rat study on key phenotypic consequences from parental (F0) high-caloric, high-fat diet (HFD) food intake, precisely and specifically at mating/conception, focusing on ‘diabesity’ risk in first- (F1) and second- (F2) generation offspring of both sexes. F0 rats (maternal or paternal, respectively) received HFD overfeeding, starting six weeks prior to mating with normally fed control rats. The maternal side F1 offspring of both sexes developed a ‘diabesity’ predisposition throughout life (obesity, hyperleptinemia, hyperglycemia, insulin resistance), while no respective alterations occurred in the paternal side F1 offspring, neither in males nor in females. Mating the maternal side F1 females with control males under standard feeding conditions led, again, to a ‘diabesity’ predisposition in the F2 generation, which, however, was less pronounced than in the F1 generation. Our observations speak in favor of the critical impact of maternal but not paternal metabolism around the time frame of reproduction for offspring metabolic health over generations. Such fundamental phenotypic observations should be carefully considered in front of detailed molecular epigenetic approaches on eventual mechanisms.

## 1. Introduction

Obesity and diabetes (‘diabesity’) are critical health challenges in westernized countries, globally, while genuine and effective measures of primary prevention are rare. Beyond decreased physical activity, high caloric food intake belongs to the key challenges for public health efforts related to diabetes and overweight prevention [1]. Notably, overweight and overnutrition are the main diabetogenic risk factors at reproductive age, in general, while gestational overweight/obesity (>30%) and gestational diabetes mellitus (GDM; >10%) are reaching epidemic prevalence [2,3,4]. Interestingly, numbers of clinical and experimental studies have shown that a predisposition to develop ‘diabesity’ may be ‘programed’ through exposure to diabetic and affluent developmental conditions already in utero and early life [5,6,7]. In particular, maternal obesity and accompanying gestational diabetes have been identified as respective risk factors for the offspring [8,9,10], mechanistically realized through early micro-structural and epigenetic ‘malconditioning’ [11,12].

Surprisingly, and widely reflected in recent years, is the observation of the offspring of obese fathers with affluent eating patterns and overnutrition having an increased offspring ‘diabesity’ disposition [13,14,15]. Epigenetic mechanisms are suggested as causative and are a matter of intense research efforts. In the maternal as well as the paternal side F2 offspring of the dietary-induced obese F0 generation, an increased ‘diabesity’ predisposition was reported to occur [13,16,17,18], seemingly speaking in favor for an epigenetic transgenerational transmission of acquired ‘diabesity’.

Meanwhile, it appears to be established that both maternal as well as paternal obesity, acquired through affluent eating patterns at reproductive age, are epigenetic risk factors for the offspring. This indicates a wide range of potential preventive measures and recommendations. Accordingly, a number of narrative reviews and papers proposed epigenetic transmission of acquired ‘diabesity’ over generations in both lines (maternal and paternal) [19,20,21,22], and yet this fundamental suggestion appears to be generally accepted.

However, in our opinion, various questions remain open here to primarily establish respective phenotypic effects, before suggested epigenetic mechanisms can meaningfully be explored. Especially, the time point and duration of high-fat diet (HFD) exposure, the quantity and quality of HFD diet considering its translational validity, consideration of gestational aspects for inter- and/or transgenerational consequences, and comparative estimation of sex differences in the offspring’s vulnerability appear to need a distinct reflection.

In order to contribute to a better-differentiated picture of this highly important but complex issue, we performed a large-scale approach in the rat on the phenotypic consequences of parental HFD exposure, specifically at the reproductive period (including: pre-mating, mating, conception, gestation and weaning). Our design aimed to explore whether or not key aspects of ‘diabesity’ disposition occur in the maternal vs. paternal line offspring of HFD-exposed parent/grandparent generations, under comparative consideration of the offspring’s sex.

## 2. Materials and Methods

### 2.1. Animal Model and Study Design

The experimental protocols were approved by the local animal welfare committee (G 0093/02; Lageso Berlin, Germany) and performed in accordance with the European Communities Council Directive (86/609/EEC). The animals used in these experiments were obtained from Charles River Laboratories (Sulzfeld, Germany). Rats were housed under standard conditions at 22 ± 2 °C and maintained on a 12/12 h inverse light–dark rhythm. Animals were given free access to water and food.

**Effect of maternal HFD on F1 and F2 offspring.** Female outbred Wistar rats aged 120–130 days were randomly assigned to two groups paired for body weight and were exposed to either standard chow diet (Controls, C; *n* = 7, 13.0 MJ/kg, energy 9% fat, 33% protein, 58% carbohydrates, ssniff R/M-H, Soest, Germany, Code V1534-000) or high-fat diet (HFD; *n* = 5, 17.2 MJ/kg, energy 34% fat, 23% protein, 43% carbohydrates, specific diet, Code 132006; Altromin, Lage, Germany). HFD was a modified version of the diet initially described by Levin et al. and has previously been shown as highly palatable [23,24]. F0 females were exposed to chow (F0c) or HFD (F0m) for six weeks before they were mated with normally fed males aged 150 days from the same source in a 2 to 1 ratio. Control and HFD dams were kept on their respective diets throughout gestation and lactation. Body weight was recorded weekly. Dams were individually housed and allowed to deliver spontaneously. Litters of F1 offspring were not culled or adjusted. At weaning (day 21 of life), half of the male and female pups from chow-fed (F1c, *n* = 36) or HFD-fed (F1m, *n* = 35) dams were randomly selected for determination of metabolic parameters and body fat. The remaining F1 pups (F1c, *n* = 44; F1m, *n* = 41) were weaned onto a standard chow diet. At 3–5 months of life, half of the adult F1 offspring of both groups and sexes (F1c, *n* = 26; F1m, *n* = 17) were sacrificed by rapid decapitation for phenotypical analyses. To generate an F2 generation, the remaining F1 female offspring of the control (F1c, *n* = 7) and HFD groups (F1m, *n* = 12) were mated with same-aged control males (F1c). All F1 dams were maintained on a standard chow diet throughout mating, gestation and lactation. Following spontaneous delivery, newborn F2 litters were not culled or adjusted. Exactly as for F1, half of the F2 offspring (F2c, *n* = 30; F2m, *n* = 73) were randomly culled for metabolic investigations on day 21 of life. The remaining F2 pups of both groups and sexes were weaned onto a standard chow diet and sacrificed at adulthood (3–5 months of age) by rapid decapitation (F2c, *n* = 25; F2m, *n* = 27). Blood and tissues were collected and carcasses were kept for body composition measurements. A schematic of the overall study design is shown in Figure 1.

**Effect of paternal HFD diet on F1 offspring.** Male outbred Wistar rats aged 100–130 days were randomly divided into two groups of equal average body weight and fed either standard chow (Controls, F0c, *n* = 7) or a high-fat diet (HFD, F0p, *n* = 10). F0 male founders were mated after six weeks of diet with chow-fed, same-aged females from the same source (Figure 1). Each male rat was used only once for mating. Throughout the mating, gestation and lactation period, rearing control females were singly housed and consumed only a chow diet. The F1 pups were weaned from mothers at three weeks of age onto standard chow. At weaning, half of the male and female pups from chow-fed (F1c, *n* = 49) or HFD-fed (F1p, *n* = 47) fathers were sacrificed for metabolic testing and analyses of body composition. At adult age (3–5 months), all F1 offspring were sacrificed by rapid decapitation and blood samples and tissue samples were collected (F1c, *n* = 20; F1p, *n* = 20). Note that all animals except F0 dams (F0m) and F0 male founders (F0p) were maintained on a standard chow diet for the period of the experiment (Figure 1).

### 2.2. Body Weight, Food Intake, and Body Composition

The body weight of F1 and F2 offspring was monitored and recorded throughout life. Mean food intake (MFI) of standard laboratory chow was studied beginning around day 60 of life for 30 consecutive days, with individual housing. Finally, body fat content was evaluated after sacrifice by drying the carcass mass (minus the stomach and intestine) to a constant weight, followed by a whole-body chloroform extraction in a Soxhlet apparatus [25]. Body fat was calculated as percentage of carcass mass.

### 2.3. Intraperitoneal Glucose Tolerance Test (IPGTT)

The IPGTT was performed in F0 dams and F0 male founders after five weeks on the respective diets, i.e., before mating. IPGTT was also performed in all F1 and F2 offspring around the age of two–three months. Following an overnight fasting period, blood was collected. Animals received an intraperitoneal injection of a 20% glucose solution (1.5 g/kg body weight). Further blood samples were taken at 15, 30, and 90 min after glucose load for determination of blood glucose levels. Using these values, the area under the curve of glucose (AUC) against time was calculated for each animal [26].

### 2.4. Metabolic Parameters

Commercially available radioimmunoassay kits (rat insulin/leptin RIA kit, Linco, St. Charles, MO, USA) were used to quantify plasma insulin and leptin concentrations. Recombinant rat insulin and leptin (Linco) served as standard preparation. The intra- and inter-assay variations for insulin were 1.4–4.6% and 8.5–9.4%, respectively, in a concentration range of 0.5–3.7 ng/mL. For leptin, the intra- and interassay variations were 2.4–4.6% and 4.8–5.7%, respectively, in a concentration range of 1.6–11.6 ng/mL. Blood glucose was measured photometrically using the glucose oxidase–peroxidase (GOD-PAP) method (Dr Lange GmbH, Berlin, Germany). The assays were performed according to manufacturer’s protocols. As an indicator of insulin resistance, the homeostatic model assessment of insulin resistance (HOMA-IR) was calculated according to Matthews et al. [27,28].

### 2.5. Statistical Analyses

Data are expressed as means ± SEM. Group differences were analyzed by Student’s *t*-test after verifying that the data were normally distributed. Otherwise, the Mann–Whitney U-test was used. To compare frequencies (e.g., on fertility) between groups, the Chi-squared test was used. Statistical analyses were calculated with GraphPad Prism Version 7.00 (GraphPad Software, Inc., San Diego, CA, USA) and SPSS 23.0 software (IBM, Munich, Germany). Statistical significance was set at *p* < 0.05.

## 3. Results

### 3.1. Metabolic Profile of F0 Dams

Around conception, i.e., after six weeks on respective diets, HFD-overfed dams (F0m) showed significantly higher weight gain than control dams (F0c) fed a chow diet (+92%, F0c: 16.5 ± 2.3 g vs. F0m: 31.7 ± 2.9 g, *p* = 0.002, Figure 2A). Before mating, rats underwent an IPGTT. F0 females on HFD showed markedly increased fasting glucose (F0c: 3.7 ± 0.1 mmol/L vs. F0m: 4.2 ± 0.2 mmol/L, *p* = 0.039), AUC during IPGTT (F0c: 18.9 ± 0.6 mmol/L/h vs. F0m: 23.1 ± 1.4 mmol/L/h, *p* = 0.011), plasma insulin (F0c: 0.2 ± 0.04 ng/mL vs. F0m: 1.0 ± 0.2 ng/mL, *p* = 0.003), and HOMA-IR (F0c: 0.7 ± 0.1 vs. F0m: 3.9 ± 0.8, *p* = 0.003) compared to chow-fed dams, indicating glucose intolerance and insulin resistance in the overfed dams. In addition, the adiposity marker leptin was clearly increased in HFD dams (F0c: 1.6 ± 0.2 ng/mL vs. F0m: 2.6 ± 0.3 ng/mL, *p* = 0.008; Figure 2A). High-fat diet consumption during the premating period did not significantly affect fertility of dams (F0c: 88% (7 of 8), F0m: 50% (5 of 10); *p* = 0.240).

### 3.2. Effect of Maternal HFD-Overfeeding on F1 Offspring

Maternal HFD overfeeding had no significant effect on the average litter size of the F1 generation (F1c: 12.7 ± 2.6 vs. F1m: 15.4 ± 2.6, *p* = 0.111). Perinatal mortality (day 0–21 of life) was even higher in the offspring of control dams (F1c: 10.0% (9 of 89)) as compared to the offspring of HFD-overfed dams (F1m: 1.3% (1 of 77), *p* = 0.040).

At weaning, the male and female offspring of dams fed an HFD showed no difference in absolute body weight as compared to controls. However, body fat content had nearly doubled in the HFD offspring (males: +95%, F1c: 6.4 ± 0.2% vs. F1m: 12.5 ± 0.2%, *p* < 0.001; females: +85% F1c: 7.1 ± 0.2% vs. F1m: 13.1 ± 0.3%, *p* < 0.001) as compared to the offspring of control dams (Figure 2B,C). Obesity of HFD offspring was associated with pronounced hyperleptinemia (males: F1c: 1.3 ± 0.1 ng/mL vs. F1m: 4.8 ± 0.3 ng/mL, *p* < 0.001; females: F1c: 1.6 ± 0.1 ng/mL vs. F1m: 4.4 ± 0.3 ng/mL, *p* < 0.001) and hyperinsulinemia (males: F1c: 0.6 ± 0.1 ng/mL vs. F1m: 0.8 ± 0.1 ng/mL, *p* = 0.042, females: F1c: 0.6 ± 0.1 ng/mL vs. F1m: 1.0 ± 0.1 ng/mL, *p* < 0.001). Furthermore, HFD offspring showed increased HOMA-IR compared with their corresponding controls, indicating early-onset insulin resistance (males: F1c: 4.2 ± 0.3 vs. F1m: 6.2 ± 0.8, *p* = 0.019; females: F1c: 3.9 ± 0.3 vs. F1m: 7.6 ± 0.5, *p* < 0.001, Figure 2B,C).

The metabolic phenotype observed at weaning persisted into adulthood. Adult male and female F1 offspring of HFD-exposed dams were clearly obese, characterized by increased body fat content (males: F1c: 13.6 ± 0.8% vs. F1m: 19.3 ± 1.1%, *p* < 0.001; females: F1c: 10.6 ± 0.5% vs. F1m: 13.9 ± 0.7, *p* = 0.001). Consistent with increased adiposity, HFD offspring presented higher plasma leptin levels as compared to the offspring of the chow-fed dams (males: F1c: 8.8 ± 0.8 ng/mL vs. F1m: 14.7 ± 1.2 ng/mL, *p* < 0.001; females: F1c: 3.4 ± 0.5 ng/mL vs. F1m: 5.4 ± 0.6 ng/mL, *p* = 0.010, Figure 2B,C). Remarkably, as at weaning, throughout life, absolute body weight did not differ between groups. To evaluate the impact of maternal high-fat overfeeding on glucose homeostasis in the offspring, IPGTT was performed at adult age. In offspring born to HFD-fed dams, the glucose area under the curve was significantly elevated (males: F1c: 16.5 ± 0.5 mmol/L/h vs. F1m: 18.8 ± 0.7 mmol/L/h, *p* = 0.006; females: F1c: 15.6 ± 0.5 mmol/L/h vs. F1m: 18.7 ± 0.6 mmol/L/h, *p* < 0.001), indicating reduced glucose uptake. Accordingly, HFD offspring showed increased insulin resistance as reflected by increased HOMA-IR (males: F1c: 15.9 ± 2.0 vs. F1m: 48.6 ± 10.4, *p* = 0.002; females: F1c: 5.9 ± 0.8 vs. F1m: 10.5 ± 1.4, *p* = 0.002), and higher basal insulin levels as compared to offspring of the control group (males: F1c: 3.2 ± 0.4 ng/mL vs. F1m: 8.6 ± 1.7 ng/mL, *p* = 0.003; females: F1c: 1.3 ± 0.2 ng/mL vs. F1m: 2.2 ± 0.2, *p* = 0.004); both alterations were more pronounced in males (Figure 2B,C). No significant differences in mean food intake were observed among the groups, either in males (F1c: 30.2 ± 0.4 g/d vs. F1m: 29.9 ± 0.5 g/d, *p* = 0.656) or in females (F1c: 21.9 ± 0.3 g/d vs. F1m: 21.1 ± 0.3 g/d, *p* = 0.063; Figure 2B,C).

### 3.3. Metabolic Profile of F0 Male Founders

Over the six-week premating period, male rats on the HFD gained significantly more weight than chow-fed control males (+41%; F0c: 74.8 ± 6.7 g vs. F0p: 105.4 ± 4.3 g, *p* = 0.001), resulting in an increased body weight (+5%; F0c: 510 ± 6 g vs. F0p: 535 ± 4 g, *p* = 0.003; Figure 3A). This was accompanied by elevated plasma leptin levels (+82%; F0c: 6.5 ± 0.6 ng/mL vs. F0p: 11.8 ± 1.0 ng/mL, *p* = 0.001), as an index of increased adiposity. Furthermore, during IPGTT, male founders on the HFD displayed increased blood glucose (+28%; F0c: 3.7 ± 0.1 mmol/L vs. F0p: 4.7 ± 0.1 mmol/L, *p* < 0.001), HOMA-IR (+92%; F0c: 4.5 ± 1.2 vs. F0p: 8.6 ± 0.8, *p* = 0.011), AUC (+24%; F0c: 20.8 ± 0.3 mmol/L/h vs. F0p: 25.7 ± 0.4 mmol/L/h, *p* < 0.001), and insulin levels (+55%; F0c: 1.3 ± 0.3 ng/mL vs. F0p: 2.0 ± 0.2 ng/mL, *p* = 0.062) as compared to control males (Figure 3A).

After the F0 male founders were exposed to an HFD for six weeks, they were mated with chow-fed control females. To ensure that there were no differences between female dams, body weight and metabolic parameters were evaluated and IPGTT was performed before mating. Females serving as dams in both groups did not differ in body weight, plasma insulin, plasma leptin, blood glucose or glucose tolerance (data not shown); i.e., mother rats of F1p were metabolically healthy. The HFD on F0 founders did not affect fertility, as it was 100% in both groups (F0c: 7 of 7, F0p: 10 of 10; *p* = 1.000).

### 3.4. Effect of Paternal HFD-Overfeeding on F1 Offspring

There were no significant differences in the average litter size at birth (F1c: 11.6 ± 5.1, vs. F1p: 12.4 ± 3.6; *p* = 0.664) or perinatal mortality (day 0–21 of life) between the offspring of chow-fed founders (F1c: 0.8% (1 of 127)) and the offspring of HFD-overfed founders (F1p: 1.6% (2 of 124) *p* = 0.983).

Paternal HFD exposure had no effect on offspring’s body weight, body fat or metabolic parameters at weaning, either in males or in females (Figure 3B,C). Remarkably, also in adulthood, male and female offspring showed no differences in absolute body weight (males: F1c: 477 ± 10 g vs. F1p: 488 ± 17g, *p* = 0.560; females: F1c: 271 ± 5 g vs. F1p: 280 ± 7 g; *p* = 0.285) and body fat (males: F1c: 13.1 ± 0.6 g vs. F1p: 13.1 ± 0.9 g, *p* = 0.985; females: F1c: 12.8 ± 0.8 g vs. F1p: 12.9 ± 0.6 g; *p* = 0.946). The offspring of both groups and sexes did not differ in any of the metabolic parameters investigated, nor in mean food intake. Furthermore, during IPGTT at three months of age, AUC were unchanged (males: F1c: 9.5 ± 0.2 mmol/L/h vs. F1p: 10.2 ± 0.4 mmol/L/h, *p* = 0.154; females: F1c: 9.4 ± 0.3 vs. F1p: 9.3 ± 0.2, *p* = 0.832), as were HOMA levels (males: F1c: 20.9 ± 2.6 vs. F1p: 18.7 ± 2.5, *p* = 0.544; females: F1c: 10.2 ± 1.7 vs. F1p: 8.4 ± 0.9, *p* = 0.529), showing that paternal HFD exposure did not affect glucose tolerance or insulin sensitivity in the later life of offspring (Figure 3B,C).

### 3.5. Metabolic Profile of F1 Dams

At the age of three months, F1 female offspring of the control (F1c) and HFD group (F1m) were mated with control males to produce F2 offspring. Around conception, no differences in body weight were observed between F1c and F1m females but adiposity marker plasma leptin was significantly increased in F1m females compared with their counterparts (F1c: 2.1 ± 0.1 ng/mL vs. F1m: 3.8 ± 0.3 ng/mL, *p* < 0.001, Figure 4A).

Although there was no significant difference in basal blood glucose level between F1c and F1m females (F1c: 3.8 ± 0.1 mmol/L vs. F1m: 3.7 ± 0.1 mmol/L, *p* = 0.510), F1m dams were glucose intolerant, showing higher AUC after glucose challenge than F1c dams (F1c: 15.7 ± 0.8 mmol/L/h vs. F1m: 18.8 ± 0.7 mmol/L/h, *p* = 0.010). Moreover, levels of insulin were significantly higher in F1m dams (F1c: 0.2 ± 0.04 ng/mL vs. F1m: 0.4 ± 0.1 ng/mL, *p* = 0.012), accompanied by elevated HOMA-IR (F1c: 0.6 ± 0.1 vs. F1m: 1.3 ± 0.2, *p* = 0.013), indicating a state of insulin resistance (Figure 4A). Fertility of F1 dams did not significantly differ between groups (F1c: 70% (7 of 10), F1m: 92% (12 of 13), *p* = 0.399).

### 3.6. Effect of Maternal HFD-Overfeeding on F2 Offspring

In the maternal-side F2 offspring of F0 dams, no significant differences in average litter size at birth (F2c: 11.6 ± 3.8 vs. F2m: 13.0 ± 5.4; *p* = 0.545) or perinatal mortality (day 0–21 of life) between the offspring of the control group (F2c: 16% (13 of 81) and HFD offspring (F2m: 11.5% (18 of 156), *p* = 0.439) were observed.

At weaning on day 21 of life, offspring of both groups and sexes did not differ in body weight, but male F2m offspring showed obesity, characterized by increased body fat content (+11%; F2c: 8.7 ± 0.3% vs. F2m: 9.6 ± 0.3%, *p* = 0.034) and elevated plasma leptin levels (+27%; F2c: 2.6 ± 0.2 ng/mL vs. F2m: 3.3 ± 0.2 ng/mL, *p* = 0.043) as compared to controls. The phenotype observed in the F2m males continued into adulthood, with adiposity and hyperleptinemia even more pronounced in later life (body fat: +24%; F2c: 13.4 ± 0.8% vs. F2m: 16.6 ± 0.6%; *p* = 0.003; plasma leptin: +40%; F2c: 7.5 ± 0.7 ng/mL vs. F2m: 10.5 ± 0.6 ng/mL; *p* = 0.002, Figure 4B).

Female F2m offspring also showed increased leptin levels at weaning as compared with the corresponding controls (+25%; F2c: 2.9 ± 0.2 ng/mL vs. F2m: 3.7 ± 0.2 ng/mL, *p* = 0.014). This was accompanied with weak but significant hyperglycemia (F2c: 7.3 ± 0.1 ng/mL vs. F2m: 7.6 ± 0.1 ng/mL, *p* = 0.034). However, in females, no group differences were observed in adulthood (Figure 4C). Furthermore, no effects were observed regarding glucose tolerance (AUC) and insulin sensitivity (HOMA) at weaning or in adulthood, either in the males or in the females (Figure 4B,C).

## 4. Discussion

Focusing on the essential outcome, namely, the overall phenotype, we proved in a large-scale inter- and transgenerational rat study the widely reflected hypothesis of perinatally acquired ‘diabesity’ predisposition in the offspring generations after parental (maternal vs. paternal) high-caloric, high-fat diet (HFD) exposure at reproduction. In essence, maternal HFD exposure gave rise to increased risk in offspring generations while, in contrast, no increased ‘diabesity’ risk was observed in the offspring due to paternal HFD exposure.

HFD exposure starting six weeks before mating, i.e., around conception, did lead immediately to pronounced weight gain in F0 dams/mother rats, accompanied by hyperleptinemia, hyperinsulinemia, insulin resistance, and hyperglycemia at mating. In consequence, F1m offspring developed a clear disposition to ‘diabesity’, with increased body fat, hyperleptinemia, hyperinsulinemia, insulin resistance, and hyperglycemia both at weaning and in later life, affecting both male and female offspring. Male F1 offspring appeared to be more affected than the females. Data complement and add precision to some comparable observations [29,30,31,32,33,34,35]. It is important to note that our periconceptional investigations clearly indicate metabolic alterations in HFD-exposed F0 mothers in terms of gestational diabetes. Therefore, data appear to support, from a mechanistically different point of view, a variety of studies reporting increased ‘diabesity’ disposition resulting from in-utero exposure to GDM [36,37]. GDM is a critical consequence of overnutrition and overweight, respectively, both clinically and experimentally [8,10,38,39]. However, it remains open whether HFD exposure itself and/or the resulting GDM are responsible for offspring alterations. Respective gestational data and/or differentiated interpretations have rarely been considered in experimental studies on parental HFD effects. Interestingly, however, even recent clinical data appear to speak in favor of the critical impact of gestational hyperglycemia here [40], instead of overnutrition/overweight per se. Accordingly, this remains to be established, with priority both clinically and in experimental designs, since the practical implications would be important.

In analogy to the F0/F1m approach, HFD exposure at mating of F0 males/founders led to hyperleptinemia, hyperinsulinemia, insulin resistance, and hyperglycemia. Here, respective control mothers were normal weight and metabolically healthy around conception and in gestation. Accordingly, but in clear contrast to F1m offspring, no obesity and/or diabetic alterations were observed in the F1p offspring, neither in early nor later life, and neither in male nor female offspring. This contrasts with observations and wide reflections on epigenetic ‘diabesity’ transmission through the paternal line [13,14,15]. Similarly, in contrast to the aforementioned studies and a variety of rather narrative reviews [20,41,42], some other similar approaches also did not confirm respective ‘diabesogenic’ alterations in the paternal line F1 offspring after HFD exposure [35,43,44]. Accordingly, data and interpretations of paternal/epigenetic ‘diabesity’ transmission appear to remain inconclusive and need further distinct exploration, as provided here. Clear characterization of the phenotypic effects appears to remain a serious prerequisite for respective epigenetic studies on eventual mechanisms.

Considering that no effect on offspring resulted from paternal, but clearly was from maternal, HFD exposure at mating, we finally focused on exploring potential effects on the maternal-side F2 offspring. Female F1m rats, shown to be affected in terms of ‘diabesity’ pre-disposition, were mated under standard conditions and with free access to normal standard chow diet with normal control males. They showed, spontaneously, hyperleptinemia, hyperinsulinemia, insulin resistance and reduced glucose tolerance around gestation. However, this was not as pronounced as in their HFD-exposed F0 mothers, as similarly described in one comparable study [17]. In consequence, few significant ‘diabesogenic’ alterations were observed in the resulting F2 (obesity, hyperleptinemia, hyperglycemia); these were more pronounced in male as compared to female offspring, with less persistence into later life than in F1m offspring. This observation appears principally to be in line with some former studies on maternal-side F2 offspring in the HFD approach [13,16,17,18], and in GDM offspring [36]. Considering the latter, especially, decreased altered gestational metabolic alterations of F1 dams as compared to their F0 mothers might provide a critical explanation for the reduced affected outcome in F2 offspring.

As customary, limitations of our approach and specific explanatory aspects must be critically discussed here, especially concerning the rather unexpected results obtained for the F1p generation.

Firstly, we applied a clear high-caloric/high-fat diet, which, when compared to most other experimental studies, is a rather moderate HFD. In various comparable investigations, a much more experimental than translationally relevant 60% high-fat diet was provided, corresponding to a 6–10-fold fat increase as compared to controls [33,45,46,47]. This appears very artificial, especially in relation to the human situation, and therefore was critically discussed recently [48]. In our study, the HFD group diet was just 3–4 times higher than the required nutritional sustenance that is standard for rat health and development (applied here to controls). This is in line with critical discussions on the necessary improvement of respective rodent HFD models, to enable a closer translational validity [48]. Interestingly, however, even this rather moderate experimental HFD exposure in F0 dams gave rise to the above-described alterations in the maternal-side offspring, but not in offspring on the paternal side.

Secondly, HFD diet overfeeding was applied here just to the F0 generations, and, most notably, in a narrow time frame at/around reproduction, i.e., not for a longer lifespan period. Especially in a translational sense, this appears mandatory to consider. While the important pioneer data on paternal-side effects in humans [49,50] dealt with exposures in rather early developmental life periods (esp. around puberty/adolescence) and/or even for a longer life span, the question of immediate effects at/around reproduction/conception appears to be of particular translational and clinical importance. Respective human data, however, are missing, as are respective animal data on paternal- vs. maternal-side effects in direct comparison and considering sex-specific offspring effects. Moreover, in the majority of experimental studies on paternal-side effects, accompanying maternal/gestational data, i.e., in terms of gestational metabolism, have hardly been documented. Therefore, it appears important to note that, in our study, a pre-mating short-term (six weeks) HFD exposure on the paternal side led to paternal ‘diabesity’ at mating but was not accompanied by increased ‘diabesity’ risk in the offspring. It should be considered, however, that our data do not exclude paternal effects if HFD exposure occurs at earlier ages, i.e., during critical developmental stages (gametogenesis/adolescence) and/or for a longer time period [49,50]. Therefore, principal avoidance of parental high-caloric HFD overnutrition around conception appears to remain a safe and recommendable preventive measure for the offspring.

Thirdly, F1m females did show ‘diabesity’ when mated with normal control males; however, in turn, the F2m generation alterations were rather weak and not as pronounced as in the F1m generation. It must be considered here that mating in the F1m was under a standard chow diet and, in turn, F1m dams’ metabolic alterations in terms of GDM were not as pronounced as observed in F0 dams under HFD exposure. Therefore, observations might point to the particular mechanistic impact of altered gestational metabolism/diabetes in terms of a dose–response effect. Interestingly, this appears to be in line with clinical observations, indicating the critical mechanistic impact of gestational diabetic alterations in both normal-weight and overweight pregnant women [40]. Interpreted integratively, data therefore seem to indicate that the occurrence and degree of the offspring alterations decisively depend on the degree of maternal metabolic alterations in terms of GDM, even irrespective of the quantity and/or quality of maternal food intake.

Finally, in both F1m and F2m, male offspring were observed to be more affected in terms of acquired ‘diabesity’ disposition than females. Similar has been observed in earlier studies [13,29,30,31,32,33,34,35]. Reasons for this sex-specific divergence are unclear [51] and remain to be explored with priority. Future studies should therefore include both sexes in a comparative design.

## 5. Conclusions

In summary and conclusion, maternal high-caloric HFD overnutrition during the time frame around reproduction is a risk factor for ‘diabesity’ predisposition in the F1 as well as maternal-side F2 generations of both sexes, with male offspring more affected than females. In contrast, paternal HFD exposure at reproduction/mating could not be identified as a respective risk factor for the offspring. The data suggest alterations of maternal metabolism (adipogenic, diabetogenic) as the decisive mechanistic aspect for acquired offspring phenotypes. Interpreted translationally, preventive dietary and metabolic measures (avoidance of overnutrition/overweight around reproduction; adequate screening for/treatment of gestational diabetes) appear to remain key to the issue. Future research priorities in the epigenetic field should carefully consider sex-specific phenotypic effects from distinct dietary compounds during distinct time windows of early/perinatal development. From a wider perspective, principal public health recommendations, however, already seem important at this stage: general avoidance of affluent eating patterns around reproduction and in pregnancy, universal screening for gestational diabetes in all pregnant women, and optimal treatment of GDM to strictly ensure normoglycemia. These measures appear to have a public health preventive impact, potentially for generations.

## Figures and Tables

**Figure 1 ijerph-17-04229-f001:**
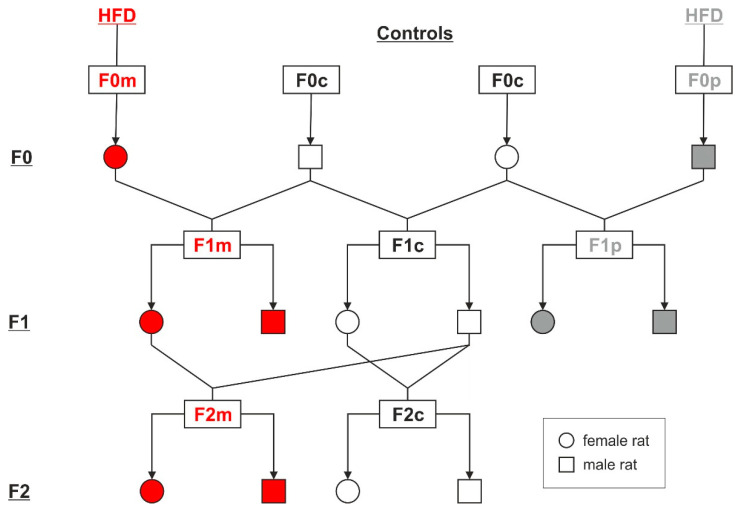
Schematic diagram of the experimental design. F0 female (F0m) and male (F0p) Wistar rats fed a high-fat diet (HFD) for six weeks were mated with respective chow-fed (F0c) control rats to generate F1 offspring. The resulting F1 progeny was fed a standard chow diet. Six-month-old female control (F1c) and HFD offspring (F1m) were then mated with same-aged control F1c males, resulting in an F2 generation (F2c, F2m).

**Figure 2 ijerph-17-04229-f002:**
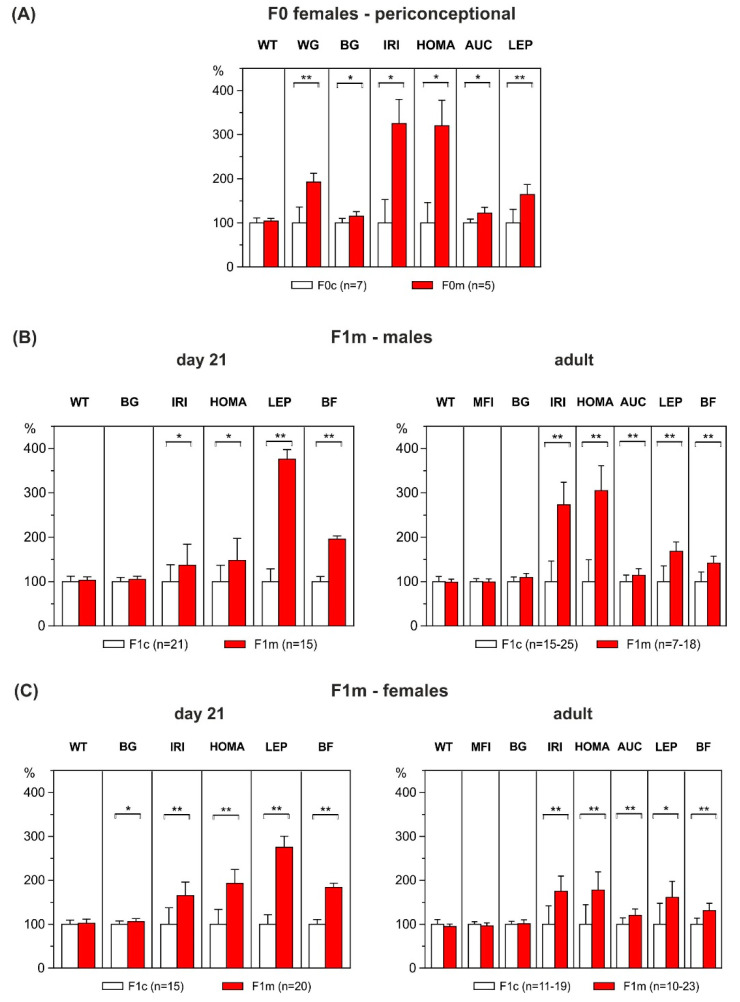
Phenotypic characteristics of HFD-overfed dams (F0m) around conception, and outcome of their male and female offspring (F1m) at weaning and adult age. (**A**) Maternal (F0m) pre-gestational weight (WT), HFD-related weight gain (WG), blood glucose (BG), plasma insulin (IRI), HOMA, area under the curve of glucose (AUC) and plasma leptin (LEP) levels after six weeks of high-fat feeding, as compared to control dams (F0c). (**B**) F1 male and (**C**) female body weight (WT), BG, IRI, HOMA, LEP, percentage of body fat (BF) and mean food intake (MFI) in HFD offspring (F1m) at weaning (day 21 of life) and at adult age, as compared to respective offspring of control dams (F1c). Data are means ± SEM, shown as percentages of control levels. *p* values were calculated using Student’s *t*-test or Mann–Whitney U-test (maternal IRI and HOMA; IRI and HOMA in male offspring, HOMA and leptin in adult female offspring) when appropriate. * *p* < 0.05, ** *p* < 0.01.

**Figure 3 ijerph-17-04229-f003:**
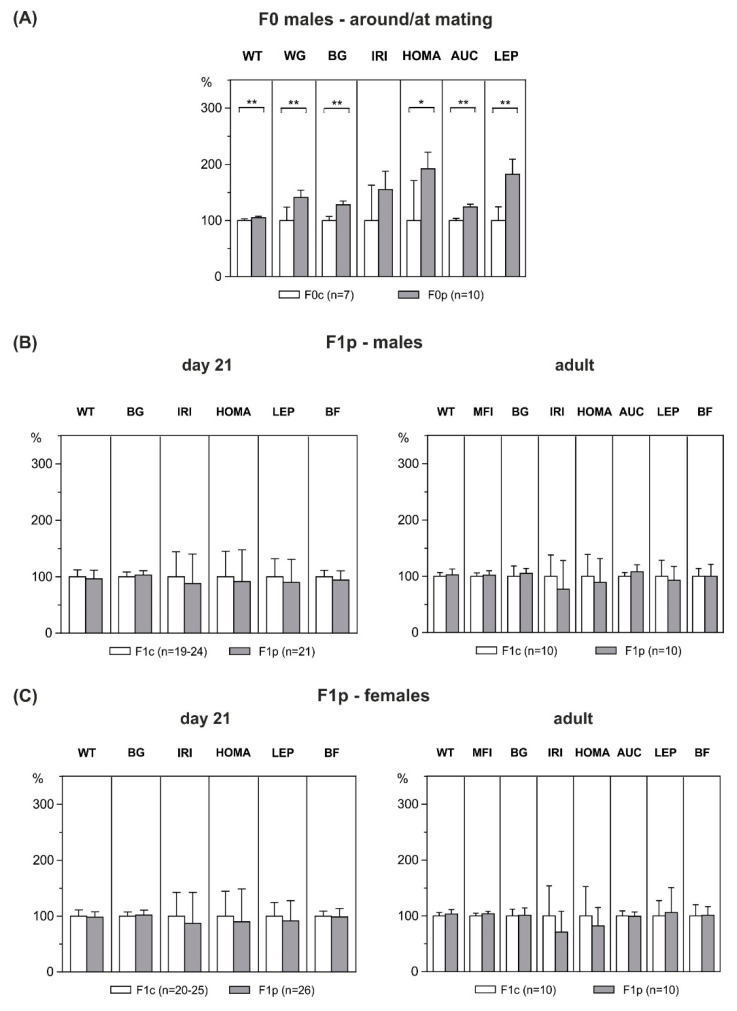
Phenotypic characteristics of HFD-overfed male founders (F0p) and outcomes of their male and female offspring (F1p) at weaning and adult age. (**A**) Paternal (F0p) pre-mating weight (WT), HFD-related weight gain (WG), blood glucose (BG), plasma insulin (IRI), HOMA, area under the curve of glucose (AUC) and plasma leptin (LEP) levels after six weeks of high-fat feeding, as compared to control founders (F0c). (**B**) F1 male and (**C**) female body weight (WT), BG, IRI, HOMA, LEP, percentage of body fat (BF) and mean food intake (MFI) in HFD offspring (F1p) at weaning (day 21 of life) and at adult age, as compared to respective offspring of control rats (F1c). Data are means ± SEM, shown as percentages of control-levels. *p* values were calculated using Student’s *t*-test or Mann–Whitney U-test (paternal leptin) when appropriate. * *p* < 0.05, ** *p* < 0.01.

**Figure 4 ijerph-17-04229-f004:**
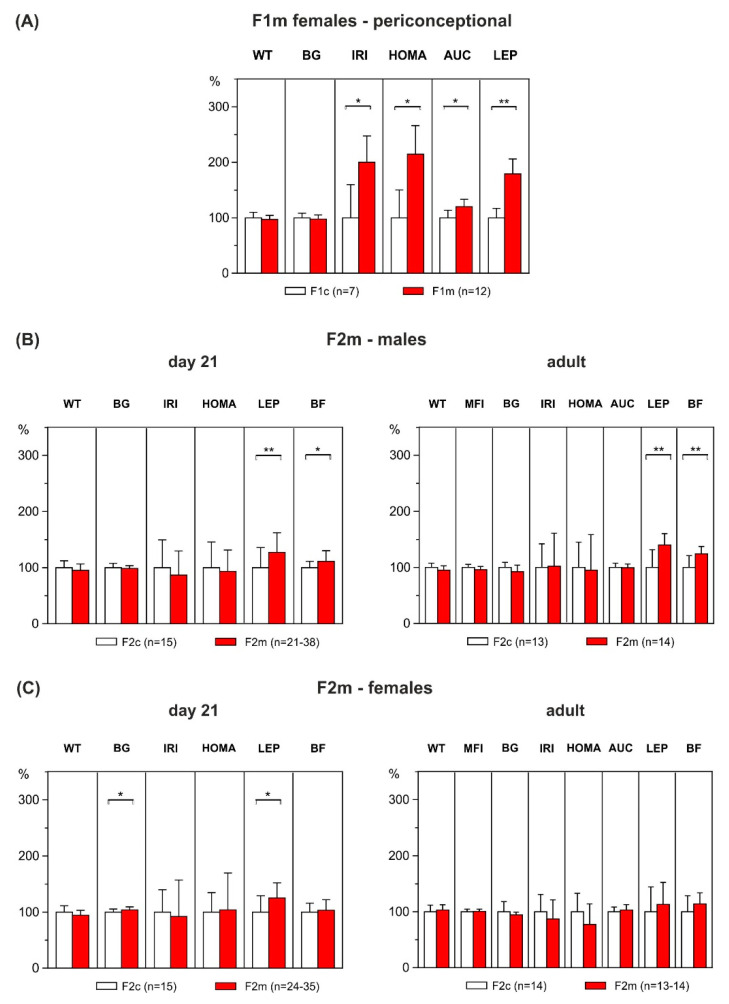
Phenotypic characteristics of F1m dams and outcome of their male and female offspring (F2m) at weaning and adult age. (**A**) Maternal (F1m) pre-gestational weight (WT), HFD-related weight gain (WG), blood glucose (BG), plasma insulin (IRI), HOMA, area under the curve of glucose (AUC) and plasma leptin (LEP) as compared to control dams (F1c). (**B**) F2 male and (**C**) female body weight (WT), BG, IRI, HOMA, LEP, percentage of body fat (BF), mean food intake (MFI) in HFD-offspring (F2m) at weaning (day 21 of life) and at adult age, as compared to respective offspring of control dams (F2c). Data are means ± SEM, shown as percentages of control-levels. *p* values were calculated using Student’s *t*-test or Mann–Whitney U-test (maternal HOMA) when appropriate. * *p* < 0.05, ** *p* < 0.01.

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
