# Peer review of "Maternal but Not Paternal High-Fat Diet (HFD) Exposure at Conception Predisposes for ‘Diabesity’ in Offspring Generations"

_ijerph, 2020, doi:10.3390/ijerph17124229_

Round 1
Reviewer 1 Report
Overall this is a valuable and well written paper. Given the scope of the target journal, I would like to see a clearer link between the maternal dietary environment and public health protection. This could be exapanded through the introduction and discussion. Currently the paper presents an narrow (single discipline) interpretation of findings. It is common for animal studies to loose sight of the broader impetus for the research.
The results are text heavy and may benefit from the inclusion of table/s to improve readability.
A thorough interpretation of the findings has been presented, in the context of animal studies. This is a well designed study and well written paper, however it would benefit by considering public health priorities and implications. The 'where to from here' is lacking. How can these findings be interpreted in consideration of future research and practice priorities.
Reviewer 2 Report
I consider the paper very clear and very interesting. The results are hard to go along (a lot of numbers in the text), but I have not an alternative. The solution is to read carefully.
I have not suggestions for changes. Congratulations.
Reviewer 3 Report
A well-executed animal model for diabesity. And a well-written manuscript. There are some minor colloquial phrases used in the discussion section. Otherwise, this manuscript is overall an excellent document.
